# Water-Level Fluctuation Control of the Trophic Structure of a Yangtze River Oxbow

**DOI:** 10.3390/biology12101359

**Published:** 2023-10-23

**Authors:** Longhui Qiu, Fenfen Ji, Yuhui Qiu, Hongyu Xie, Guangyu Li, Jianzhong Shen

**Affiliations:** 1Engineering Research Center of Green Development for Conventional Aquatic Biological Industry in the Yangtze River Economic Belt, Ministry of Education, College of Fisheries, Huazhong Agricultural University, Wuhan 430070, China; longhui@webmail.hzau.edu.cn (L.Q.);; 2Fisheries College, Jimei University, Xiamen 361021, China

**Keywords:** Tian-e-Zhou Oxbow, water-level fluctuation, trophic structure, wet season, dry season, Yangtze finless porpoise

## Abstract

**Simple Summary:**

Seasonal fluctuations in the water level have the potential to alter the nutrient composition of the aquatic ecosystem and disrupt the trophic structure. Tian-e-Zhou Oxbow is the first ex situ reserve established for the Yangtze finless porpoise in China. The extant rich and abundant forage fish resources are a guarantee for the healthy development of the Yangtze finless porpoise population. However, the water-level difference between the Tian-e-Zhou Oxbow and the Yangtze mainstream has been increasing due to the barrier of the Sha-Tan Levee, and effective regulation measures have not been implemented. In this study, we aimed to explore the impact of water-level fluctuations on the trophic structure of the ecosystem in the Tian-e-Zhou Oxbow by analyzing the trophic position, trophic niche, and resource contribution to fish during both the wet and the dry seasons. The study will contribute to the conservation of the Yangtze finless porpoise and provide a valuable reference for habitat improvement and restoration efforts in the Tian-e-Zhou Oxbow of the Yangtze River.

**Abstract:**

Seasonal water-level fluctuations can profoundly impact nutrient dynamics in aquatic ecosystems, influencing trophic structures and overall ecosystem functions. The Tian-e-Zhou Oxbow of the Yangtze River is China’s first ex situ reserve and the world’s first successful case of ex situ conservation for cetaceans. In order to better protect the Yangtze finless porpoise, the effects of water-level fluctuations on the trophic structure in this oxbow cannot be ignored. Therefore, we employed stable isotope analysis to investigate the changes in the trophic position, trophic niche, and contribution of basal food sources to fish during the wet and dry seasons of 2021–2022. The research results indicate that based on stable isotope analysis of the trophic levels of different dietary fish species, fish trophic levels during the wet season were generally higher than those during the dry season, but the difference was not significant (*p* > 0.05). Fish communities in the Tian-e-Zhou Oxbow exhibited broader trophic niche space and lower trophic redundancy during the wet season (*p* < 0.05), indicating a more complex and stable food web structure. In both the wet and dry seasons, fish in the oxbow primarily relied on endogenous carbon sources, but there were significant differences in the way they were utilized between the two seasons (*p* < 0.05). In light of the changes in the trophic structure of the fish during the wet and dry seasons, and to ensure the stable development of the Yangtze finless porpoise population, we recommend strengthening the connectivity between the Tian-e-Zhou Oxbow and the Yangtze River.

## 1. Introduction

Among the significant drivers of change in global freshwater aquatic communities, short-term water-level fluctuations play a prominent role [1]. These fluctuations lead to hydrological alterations that directly impact the habitats of aquatic organisms [2,3], thereby influencing the composition of aquatic communities and trophic relationships within the food chain [4,5,6] and impacting the overall stability and sustainability of the river and lake ecosystems [7,8]. Under the influence of the monsoon season, the Yangtze River basin experiences significant seasonal variations in precipitation [9], which causes periodic fluctuations in river discharge and water levels. Fish, as key components of the food web, play a crucial role in maintaining ecological balance and aquatic biodiversity, occupying both predator and prey positions [10,11]. However, studies have indicated that the species diversity of fish and other aquatic organisms in the Yangtze River is highly sensitive to hydrological changes [12,13]. Meanwhile, hydrological fluctuations within the Yangtze River basin perpetually exert an impact on the nutritional composition of aquatic ecosystems [14,15,16].

Stable isotope analysis (SIA) is a powerful and widely employed tool for examining energy flows and nutrient pathways within the food webs of various aquatic ecosystems [17,18]. This methodology enables the identification of dietary variability, feeding strategies, trophic positions, trophic niche shifts, and nutrient sources within food webs [19,20,21]. Generally, the stable isotope ratios δ^13^C and δ^15^N are utilized to determine the resource pools supporting the consumers and to establish the trophic position of organisms in the food web, respectively [22,23]. Additionally, Bayesian mixing models constructed using stable isotope ratios δ^13^C and δ^15^N allow for estimating the contributions of potential prey items and the trophic niche space occupied by consumers in the food web [24,25].

The Yangtze finless porpoise (*Neophocaena asiaeorientalis*), belonging to the Phocoenidae, is the sole freshwater population among all conspecifics. Its habitat is limited to the mainstream Yangtze River, Dongting Lake, and Poyang Lake [26]. Designated as endangered by the International Union for the Conservation of Nature (IUCN) in 1996, the Yangtze finless porpoise was elevated to critically endangered status in 2013 [27]. Moreover, in the updated 2013 List of National Key Protected Wildlife, the Yangtze finless porpoise has been reclassified from the second level to the first level of national protection. Food shortage was considered to be the main reason for the decreasing population of Yangtze finless porpoise [28]. One study analyzed the diet of 17 Yangtze finless porpoises from the Yangtze River by next-generation sequencing techniques and found that *C. brachygnathus*, *H. bleekeri* and *P. simoni* were the most common in the diet of Yangtze finless porpoises [29]. It was also verified that the Yangtze finless porpoise is an opportunistic predator like East Asian and Indo-Pacific finless porpoises, which shows how much food iss related to fish community structure [30,31,32].

The Tian-e-Zhou Oxbow, located in the Yangtze River, serves as the first ex situ reserve established in China for the conservation of the Yangtze finless porpoise. In 1991, five Yangtze finless porpoises were introduced to the Tian-e-Zhou Oxbow from the Yangtze River, marking a crucial milestone [33]. Over the years, the population of Yangtze finless porpoises in the Tian-e-Zhou Oxbow has experienced significant growth, from more than 60 in 2015 to 101 in 2021 [34,35]; this makes the Tian-e-Zhou Oxbow the world’s first successful case of ex situ conservation for cetaceans. The Père David’s deer (*Elaphurus davidianus*) nature reserve was established in 1991 and is located adjacent to the Tian-e-Zhou Oxbow. Since the introduction of 64 Père David’s deer in 1993, the current population has exceeded 2500. Rising water levels in the Tian-e-Zhou Oxbow will reduce the size of the Père David’s deer nature reserve and may threaten the survival of the Père David’s deer, so the oxbow is often operated at a low water level, with only a small amount of Yangtze River water being allowed to reach the oxbow during the wet season [36]. Moreover, the construction of the Sha-Tan Levee has had a significant impact on the hydrology of the Tian-e-Zhou Oxbow, which has led to an increase of at least 0.5 m in the water level difference between the Yangtze River channel and Tian-e-Zhou Oxbow [37], disrupting water exchange between the two ecosystems, which is detrimental to the protection and development of fish species and Yangtze finless porpoises in the Tian-e-Zhou Oxbow.

Previous studies have shown that seasonal changes in hydrology can affect the population structure of fish and other aquatic organisms in the Tian-e-Zhou Oxbow [35,38,39]. Li and Wang [40] used Ecopath with Ecosim 6.6.5 software to construct a mass balance model for the Tian-e-Zhou Oxbow in 2017; this model portrayed the fish trophic structure and estimated the environmental capacity for Yangtze finless porpoises, determining it to be 89 individuals. However, the number of Yangtze finless porpoises in the Tian-e-Zhou Oxbow reached 101 in 2021, which exceeds the environmental carrying capacity estimated by Li and Wang [40], and the current fish trophic structure is bound to change significantly from 2017. Compared with the results estimated by stable isotope analysis (SIA), the Ecopath model may underestimate the trophic level of functional groups [41]. Understanding the ecological effects of water-level fluctuation on the trophic structure in Tian-e-Zhou Oxbow is crucial for the conservation of the Yangtze finless porpoise. Therefore, stable isotope techniques were used in this study to (1) quantitatively describe changes in the trophic structure of the Tian-e-Zhou Oxbow during the wet and dry seasons, (2) analyze changes in the contributions of basal resources to consumers, and (3) provide recommendations for improving water-level management in the Tian-e-Zhou Oxbow.

## 2. Materials and Methods

### 2.1. Study Area

Located in Hubei Province, Central China, the Tian-e-Zhou Oxbow (29°46′71″–29°51′45″ N, 112°31′36″–112°37′30″ E) is a notable geographical feature formed through the natural cutting and straightening of the Yangtze River in 1972 [42]. Upon the completion of the Sha-Tan Levee in 1988, the Tian-e-Zhou Oxbow was almost completely isolated (Figure 1) [43], leaving only a narrow channel connected to the Yangtze River; water levels were mainly regulated by man-made control through a sluice gate in the middle of the channel. The sluice gate is opened during the wet season to allow a specific volume of water from the Yangtze River to enter, while it remains closed at all other times; this leads to periodic changes in water levels, as in 2021 (Figure 2). With a subtropical monsoon climate, the oxbow experiences an average annual temperature of 16.5 °C and precipitation of 1200 mm [37]. Spanning approximately 20.9 km in length, the oxbow covers a surface area of 16 km², with a mean depth of 4.5 m and a maximum depth of 25 m. Adjacent to Dawan town, Xiaohekou town, and Henggoushi towns, the total area encompassing these three towns measures 433.6 km², with a population of 128.5 thousand, predominantly engaged in farming activities. Among these towns, Dawan and Xiaohekou have no direct connection to the oxbow, whereas only Henggoushi Town is linked to the oxbow via a small aqueduct at Huanggua Village. However, this connection has minimal impact on the water-level fluctuations of the oxbow.

### 2.2. Sample Collection and Preparation

In this study, concurrent sampling of different basal sources, invertebrates, and fish was conducted in the Tian-e-Zhou Oxbow of the Yangtze River during the dry season (October) and during the wet season (April) from 2021 to 2022. In April, 47, 41, and 45 samples were collected from Xindi, Huanggua, and Zhengjiatai Villages, respectively, compared to 43, 37, and 41 in October. The collection of basal sources included phytoplankton, particulate organic matter (POM), submerged plants, emerging plants, and sedimentary organic matter (SOM). Submerged plant and emerging plant leaves were manually collected and carefully washed with distilled water to remove any detritus. Phytoplankton samples were obtained by towing a 112 µm plankton net and filtered through precalcined (450 °C for 5 h) Whatman GF/F filters. POM was collected by filtering a combined water sample obtained by mixing water collected from different depths (upper, middle, and lower layers) using a 64 µm plankton net and placed onto precalcined GF/F filters under moderate vacuum (10 mbar) until clogging occurred [44]. The top 0–1 cm layer of surface sediment was collected using a Peterson grab and promptly transferred into sterile plastic bags [45].

The consumer organisms in this study included zooplankton, mollusks, crustaceans, and fish. Zooplankton samples were collected by trawling a plankton net (with a mesh size of 112 µm) through the upper water layer for a duration of 5 min. The collected samples were then filtered through Whatman GF/F filters. Mollusks and crustaceans were collected using a Peterson grab and D-framed nets. Fish samples were captured using a purse net (with dimensions of 350 m long × 10 m high and a mesh size of 0.5 cm) and gillnets (with dimensions of 20 m × 1 m and a mesh size of 2.5 cm). Upon capture, the mollusks, crustaceans, and fish were immediately identified at the species level [46,47,48]. General fish characteristics, including measurements, were recorded. Tissue samples from the white muscle, located between the dorsal fin and the lateral line, were taken from the fish specimens. Skin and scales were carefully removed from the samples [49,50]. Muscle samples were promptly packed into sterile Eppendorf tubes and stored in a −20 °C refrigerator until they were transported to the laboratory for further analysis [51].

All the collected samples, including aquatic plants, phytoplankton, POM, SOM, zooplankton, mollusks, crustaceans, and fish muscle, were dried in an oven at 60 °C for 48 h until they reached a constant weight. Subsequently, the dried samples were ground into a homogeneous powder using a glass mortar and pestle, ensuring consistency in particle size [44]. To prepare the samples for δ^13^C measurement, the aquatic plants, phytoplankton, POM, and SOM underwent a carbonate removal treatment using 1 mol/L HCl. The acid was added drop by drop to the samples, effectively eliminating carbonates. The treated samples were then subjected to another round of drying at 60 °C and subsequently ground into a fine powder. The samples intended for δ^15^N analysis did not undergo any acidification treatment [44,52].

Environmental metrics were measured from April to December 2021–2022 at 5 to 6 sampling sites of the Tian-e-Zhou Oxbow. A portable YSI Professional Plus instrument was utilized for measuring water temperature (WT), pH, dissolved oxygen (DO), and total dissolved solids (TDS) (Table 1). Secchi depth (SD) was also determined by using a Secchi disk. The permanganate index (COD_Mn_) was determined using the alkaline potassium permanganate titration method (GB 11892-89, China [53]). Total nitrogen (TN) and total phosphorus (TP) were analyzed using the alkaline potassium persulfate digestion-UV spectrophotometric method and the ammonium molybdate spectrophotometric method, respectively [54], with a UV-3000 spectrophotometer (MAPADA, Shanghai, China).

### 2.3. Stable Isotope Analysis

The δ^13^C and δ^15^N values of all the collected samples were analyzed using the Vario PYRO cube and Isoprime 100 instruments from German company Elementar (Langenselbold, Germany) at the public laboratory platform of the College of Resources and Environment, Huazhong Agricultural University. The δ values (‰) were defined as the ratios of stable isotopes, and the expression of this formula is as follows:
*δX* (‰) = [(*R_sample_* − *R_Standard_*)/*R_Standard_*] × 1000(1)
where *X* represents ^13^C or ^15^N, and *R* represents ^13^C/^12^C or ^15^N/^14^N. Standard samples (USGS24 and USG25) were inserted every 15 samples. The analytical precision was within 0.1‰ and 0.2‰ for the carbon and nitrogen isotope measurements, respectively.

### 2.4. Data Analysis

According to Post [55], δ^15^N enrichment among different trophic levels is typically 3.4‰. The trophic levels (*TL*) were calculated for each consumer by using the following 2 equations:*TL* = (δ^15^N*_consumer_* − δ^15^N*_baseline_*)/3.4 + *λ*(2)
where δ^15^N*_consumer_* and δ^15^N*_baseline_* are the stable isotope values of the consumer and baseline organism, respectively. The trophic fractionation of δ^15^N per trophic level is 3.4, and *λ* is the trophic level of the baseline organism [55]. *Bellamya aeruginosa*—which belongs to the family Viliparidae within the class Gastropoda of the phylum Mollusca—and was assumed to exclusively feed on simple basal resources, was chosen as the baseline organism, and λ was attributed as trophic level 2.

We utilized trophic niche metrics proposed by Layman et al. [56] to evaluate the trophic niche characteristics of fish in the Tian-e-Zhou Oxbow. The δ^15^N range (NR) quantifies the span between the most depleted and enriched δ^15^N values among species, indicating the length of the food chain. The δ^13^C range (CR) represents the range between the most depleted and enriched δ^13^C values among species, providing insights into the diversity of food sources. The total area of the convex hull (TA) and corrected standard ellipse area (SEAc) are indicative of the overall niche space and core niche space that species occupy, respectively, and they serve as proxies for measuring niche breadth. The mean distance to the centroid (CD) calculates the average Euclidean distance of each species to the centroid, offering insights into the average level of trophic diversity. The mean nearest neighbor distance (MNND) quantifies the average Euclidean distance to the nearest neighbor of each species, serving as an indicator of the overall density of species packing. Conversely, the standard deviation of nearest neighbor distance (SDNND) reflects the evenness of species packing. Lower MNND and SDNND values are indicative of heightened trophic redundancy. The SEAc calculation provides a useful tool for assessing the degree of niche overlap between species; this allows for a quantitative measure of dietary similarity [57].

We used the SIAR mixing model based on the Bayesian linear equations to determine the proportional contributions of basal resources to consumers [58]. We quantitatively analyzed the overlap in stable isotope trophic niches between fish species using the SEAc (corrected standard ellipse area) overlap, with the degree of overlap signifying the proportion of SEAc’s maximum likelihood estimates shared by two species. This estimation signifies the trophic niche overlap between the two species. An overlap degree close to 0 indicates that the two ellipses are separate, while an overlap degree near 1 signifies complete overlap of the two ellipses. Consistent with previous research findings [59,60], this study defined an overlap degree exceeding 60% as a significant niche overlap.

We carried out statistical analyses using SPSS 26.0, while the SIBER and SIAR models were run in R 4.2.3 software [59]. Data calculations and processing were performed using Excel 2017, SPSS 26, and Origin 2021 software packages.

## 3. Results

### 3.1. Isotopic Values and Trophic Levels

In this study, 203 consumers and 51 basal food source samples were collected in the Tian-e-Zhou Oxbow during the wet and dry seasons and analyzed for isotopic ratios of carbon and nitrogen. The consumers comprise 30 species of fish and 5 species of invertebrates, while the basal resources encompass 5 species of terrestrial plants and 3 species of aquatic plants, along with particulate organic matter (POM), sedimentary organic matter (SOM), and phytoplankton (Figure 3). Before the analyses, the data were checked for normality and homogeneity of variance assumptions, and logarithmic transformations were performed when needed by using SPSS 26. According to the dietary habits of the fish [61], we divided the fish in the Tian-e-Zhou Oxbow of the Yangtze River into six categories: benthivores, detritivores, herbivores, omnivores, piscivores, and zooplanktivores. The stable isotope data relative to the consumer species (32 species in both the dry and wet seasons) fell within the isotopic mixing space determined by six potential sources in both seasons, indicating that the carbon sources tested were the most probable food sources for the consumers. The results of one-way ANOVA showed that the δ^15^N values of the piscivore and omnivore fish we collected were significantly higher in the wet season than in the dry season (*p <* 0.05), while no significant differences were found for δ^13^C values (*p >* 0.05). The δ^13^C values of zooplanktivore and detritivore fish were significantly lower in the wet than in the dry season (*p <* 0.05), while the δ^15^N values of the zooplanktivore fish were significantly higher in the wet than in the dry season (*p <* 0.05).

In this study, *Bellamya aeruginosa* was chosen as the baseline organism. The mean values of nitrogen stable isotopes in dry and wet seasons of *B. aeruginosa* in Tian-e-Zhou Oxbow were 11.36‰ and 11.02‰. We found that the length of the food chain in the Tian-e-Zhou Oxbow was longer in the wet season than in the dry season (Figure 4); the trophic level of fish in the wet season ranged from 1.01 to 4.31 (averaged 2.52), and the dry season ranged from 1.33 to 3.43 (averaged 2.63). However, one-way ANOVA showed that the difference in fish trophic levels between the wet and dry seasons was not significant (*p* > 0.05). Regardless of the dry or wet season, the trophic level of *Ctenopharyngodon idellus* was always the lowest, and *Coilia brachygnathus* was the highest.

### 3.2. Trophic Niche

In the Tian-e-Zhou Oxbow of the Yangtze River, fish trophic niche metrics showed notable variations between the wet and dry seasons by one-way ANOVA. Almost all of the indicators were significantly higher in the wet season than in the dry season (*p <* 0.05), as indicated by the results presented in Table 2. Particularly, the δ^15^N range (NR) exhibited higher values in both the dry and wet seasons for piscivorous species, highlighting their position at the top of the food chain. Additionally, omnivorous fish displayed significant values for the δ^13^C range (CR), mean distance to the centroid (CD), corrected standard ellipse area (SEAc), and total area of the convex hull (TA), underscoring their substantial nutritional diversity. Notably, omnivores exhibited lower mean nearest neighbor distance (MNND) and standard deviation of nearest neighbor distance (SDNND), emphasizing their enhanced functional redundancy, particularly during the dry season.

The functional ecological niche overlap analysis was conducted on fifteen fish species within the Tian-e-Zhou Oxbow of the Yangtze River. The findings revealed a moderate degree of overlap among species during the wet season (Figure 5). Specifically, *Aristichthys nobilis* and *Hypophthalmichthys molitrix* exhibited a 64% overlap in SEAc. Similarly, *Hemiculter bleekeri* overlapped with *Culter mongolicus*, *Culter oxycephaloides*, and *Rhinogobius giurinus*, with overlap percentages of 41%, 42%, and 38%, respectively. Furthermore, *Squalidus argentatus* demonstrated 39% and 42% overlaps with *C. mongolicus* and *C. oxycephaloides*, while *Pseudobrama simoni* exhibited overlaps of 55% and 40% with *Culter dabryi* and *Xenocypris davidi*, respectively. During the dry season, *Carassius auratus* exhibited moderate overlaps of 43%, 32%, and 48% in SEAc with *C. dabryi*, *Hemiculter leucisculus*, and *S. argentatus*, respectively. Notably, *C. mongolicus* and *H. leucisculus* demonstrated a high overlap of 97%. *C. auratus* also displayed overlaps with *S. argentatus* and *X. davidi*, with percentages of 52% and 38%, respectively. Additionally, *X. davidi* exhibited a 30% overlap with *P. simoni*.

### 3.3. Contributions of Basal Resources to Consumers

Among the six primary food sources examined, including filamentous green algae, aquatic vascular plants, phytoplankton, terrestrial plants, POM, and SOM, the filamentous green algae, aquatic vascular plants, and phytoplankton are considered endogenous carbon sources, terrestrial plants are considered exogenous carbon sources, while POM and SOM contain both exogenous and endogenous nutrients. According to the Bayesian mixing models analysis, the predominant carbon sources utilized by consumers in the Tian-e-Zhou Oxbow of the Yangtze River, during both the wet and dry seasons, were found to be endogenous. However, there were significant differences in the way the fish utilized these sources between the two seasons, shown in a one-way ANOVA (*p* < 0.05).

During the period of the wet season (Figure 6), the contribution of phytoplankton to invertebrates was the highest, accounting for 50.2%. Among the fish, zooplanktivore fish demonstrated the highest reliance on phytoplankton, with a contribution percentage of 60.5%. Herbivorous and omnivorous fish showed the highest contribution of SOM as a basal food source, with percentages of 39.4% and 29.6%, respectively. Overall, the contribution of the six food sources to the consumers in the Tian-e-Zhou Oxbow of the Yangtze River ranked as follows: phytoplankton > SOM > aquatic vascular plants > filamentous green algae > POM > terrestrial plants.

In the dry season, the contribution of SOM and terrestrial plants to consumers significantly decreased, while the contribution of aquatic vascular plants, filamentous green algae, and phytoplankton increased. Phytoplankton exhibited the highest contribution rate to invertebrates, accounting for 56.8%. Herbivorous fish showed a notably higher reliance on filamentous green algae, contributing 37.6%, more than other food sources. During this period, the hierarchy of basic food sources contributing to the consumers in the Tian-e-Zhou Oxbow of the Yangtze River shifted to phytoplankton > aquatic vascular plants > filamentous green algae > POM > SOM > terrestrial plants.

## 4. Discussion

Carbon and nitrogen stable isotopes are powerful tools to study the structural characteristics of food webs [63,64], and the isotopes accumulated in different species can vary greatly [65]. Stable carbon isotopes exhibit a fractionation of 0–1‰ between adjacent trophic levels, making the δ^13^C range (CR) a valuable indicator of the diversity of the consumer food sources [66]. A higher CR value indicates a wider range of food sources available to consumers [67]. In this study, the CR value in the wet season (12.47‰) was higher than that in the dry season (6.76‰); this result is similar to Lake Victoria [68]. The wet season (spring) corresponds to the breeding season for most fish species, during which they tend to exhibit increased feeding activity in preparation for spawning. Fish typically display a generalist feeding behavior, consuming a broader array of food items during this period [69]. This heightened dietary diversity may account for the observed wide range of δ^13^C in the fish during the wet season. Omnivorous fish had the highest CR values, indicating that omnivorous fish have more flexible dietary habits, which was similar to what has been found in Poyang Lake and Dongting Lake in the Yangtze River [70]. The δ^13^C values of zooplanktivore and detritivore fish were significantly higher in the dry than in the wet season (*p <* 0.05); this means that the high δ^13^C content of the basal food resource in the dry season allows the fish to enrich in ^13^C. The contributions of endogenous and exogenous carbon to the consumers were different in different hydrological periods. During the dry season, the base resources for the consumers are dominated by endogenous carbon sources such as phytoplankton, filamentous green algae, and aquatic vascular plants, which is similar to floodplain aquatic ecosystems and highland lake ecosystems [71,72]. The contribution of terrestrial carbon sources to consumers increases during the wet season. Previous studies have shown that the inundated terrestrial organic matter can be directly consumed by fish or indirectly by fish through microbial decomposition into organic debris after the water level rises [73,74]. Currently, more than 2500 Père David’s deer inhabit the nature reserve, generating a substantial amount of excrement every year. During the wet season, when the water level of the oxbow rises, this exogenous organic matter will either directly or indirectly be introduced into the Tian-e-Zhou Oxbow, where it becomes part of the fish diet. This phenomenon can be demonstrated by the decline in soil nutrient content in the Père David’s deer nature reserve during the wet season [75]. Additionally, the TN value of the Tian-e-Zhou Oxbow peaked in July in this study (Table 1); the water level also peaked in July (Figure 2), and this observation suggests that increasing water levels facilitated the influx of nutrients into the Tian-e-Zhou Oxbow.

In this study, we found that the δ^15^N values of the piscivore and omnivore fish we collected were significantly higher in the dry season than in the wet season (*p <* 0.05). This is due to the fact that during the dry season, the reduction in lake area reduces the range of fish movement and increases fish density, resulting in increased predation intensity of piscivores and omnivores [76]. During the wet season, the δ^15^N values of both the *N. taihuensis* and *C. brachygnathus* in the Tian-e-Zhou Oxbow were higher, probably due to their relatively wide range of food sources, which was consistent with the stable isotope results in the Three Gorges Reservoir and Poyang Lake of the Yangtze River in China [77,78]. *B. aeruginosa* was chosen as the baseline organism, and we found that the food chain length of the Tian-e-Zhou Oxbow was 3.30 in the wet season and decreased to 2.10 in the dry season. As we mentioned above, the reason is that periods of wet season could help exogenous organic from the Père David’s peer nature reserve to enter the Tian-e-Zhou Oxbow, and the diversity of the fish food sources affects the enrichment of δ^15^N. The range of fish trophic levels in the Ecopath model constructed by Li and Wang in 2017 was only 2.13–3.62 [40], which is considerably narrower than the ranges observed during the wet (1.01–4.31) and dry (1.33–3.43) seasons in this study. Moreover, in the Ecopath model, piscivorous fish such as snakehead (3.62), topmouth culter (3.61), and catfish (3.55) had the highest trophic level, while *C. brachygnathus* and *N. taihuensis* were neglected, which may be due to the differences caused by the division of the functional groups and the food composition of the fish during the construction of the Ecopath model. *C. brachygnathus* and *N. taihuensis*, as the dominant fish in the oxbow [35], had mean trophic levels of 4.31, 3.85, and 3.43, 3.35 during the wet and dry seasons in this study, respectively. This result compensates for the shortcomings of the Ecopath model [40].

The trophic structure is a major characteristic of ecosystems, and understanding the factors determining trophic structure is important for predicting the response of ecological dynamics and ecosystem services to future environmental change [79]. The metrics of the isotope niche enable quantitative analysis of stable isotopes in trophic niches [80]. CR, NR, TA, SEAc, and CD were used to measure the diversity of the community trophic structure, and MNND and SDNND to indicate the magnitude of community redundancy [81]. In this study, the values of CR, NR, CD, TA, SEAc, MNND, and SDNND of the fish were significantly higher in the wet season than in the dry season (*p <* 0.05). This indicates that the basal carbon source of fish was richer, the trophic level was higher, the trophic niche width was wider, and the trophic niche uniformity was higher in the wet season in the Tian-e-Zhou Oxbow of the Yangtze River. When we compared our data with data from other water bodies in the Yangtze River basin, we observed that the trophic diversity of the community was higher in the wet season in the Tian-e-Zhou Oxbow. This was evident from the greater CR, NR, TA, SEAc, and CD values in comparison to the Three Gorges Reservoir [82], the upper Yangtze River [83], Yangcheng Lake [84], and Daning River [85] (Table 3). The number of fish species collected during the wet season amounted to 27, while during the dry season, it was 28. This diversity in species richness places the oxbow ecosystem at a notably high level among these five aquatic environments. While the redundancy is at a medium level, the MNND (0.89) and SDNND (0.62) were smaller than those of the Three Gorges Reservoir and Daning River. The trophic diversity of the fish community in the dry season of the Tian-e-Zhou Oxbow decreased very significantly compared to the wet season and was lower than that of the Three Gorges Reservoir and the upper reaches of the Yangtze River, which indicates that the ecosystem’s resistance to external disturbances was increased and stability rose.

From an ecological point of view, an overlap of greater than 60% of the SEAc represents a significant overlap [59,86]. In this study, the niches of *A. nobilis* and *H. molitrix* overlapped significantly during the wet season, accounting for 67% of the area of the *H. molitrix* trophic niche. *H. bleekeri*, *P. simoni*, *C. mongolicus*, and *S. argentatus* overlapped with other fish but not significantly. In the dry season, there were more overlapping pairs of trophic niches of *C. auratus*, *C. mongolicus*, and *X. davidi*, among which *C. mongolicus* overlapped significantly with *H. leucisculus*, accounting for 97% of the niche area of *H. leucisculus*. These results suggest that *C. mongolicus* exhibits a broader trophic niche than *H. leucisculus* in the Tian-e-Zhou Oxbow, which is consistent with its ecological performance in the Yangtze River [87]. In an environment of limited resources, full competitors cannot coexist. At the end of 2017, a fishing ban policy was implemented in the Tian-e-Zhou Oxbow, leading to a continuous growth in fish resources. As the resource abundance of *C. mongolicus* has steadily increased, that of *H. leucisculus* has continuously declined [35], which may be related to its high overlap in its trophic niche during the dry season.

Tian-e-Zhou Oxbow is the first ex situ reserve established for the Yangtze finless porpoise in China; the stabilized ecosystems and trophic structure guarantee the healthy development of the Yangtze finless porpoise population. Nevertheless, water-level fluctuations exert a significant influence on the composition of fish communities within the Tian-e-Zhou Oxbow [35,88]. The fish breeding in the Yangtze River can enter the oxbow in large numbers during the wet season, while large migratory fish cannot return to the Yangtze River during the dry season due to the barrier of the sluice gates [89]. Furthermore, as highlighted earlier in this study, the impact of water-level fluctuations on the trophic niches, food sources, and trophic structure of fish within the Tian-e-Zhou Oxbow is substantial. Consequently, enhancing the connectivity between the oxbow and the Yangtze River, along with increasing water replenishment during both wet and dry seasons, will greatly benefit the stability of the water ecosystem in the oxbow and the development of the Yangtze finless porpoise population.

## 5. Conclusions

In the current study, as the first ex situ reserve established for the Yangtze finless porpoise in China, the trophic structure of Tian-e-Zhou Oxbow has garnered significant attention. We posited that water-level fluctuations could potentially influence the nutrient dynamics in the Tian-e-Zhou Oxbow. Through stable isotope analysis, we observed that during the wet season, compared to the dry season, the food web exhibits a longer trophic chain, with fish generally occupying broader ecological niches and utilizing a more diverse array of exogenous food sources. These findings hold vital implications for the management of the Tian-e-Zhou Oxbow. However, it is important to note that due to the special conservation status of the Yangtze finless porpoise, stable isotope samples were not collected from the porpoises inhabiting the oxbow during our sampling process. Furthermore, the impact of water level fluctuations on the nutrient dynamics in different zones within the oxbow warrants further investigation.

## Figures and Tables

**Figure 1 biology-12-01359-f001:**
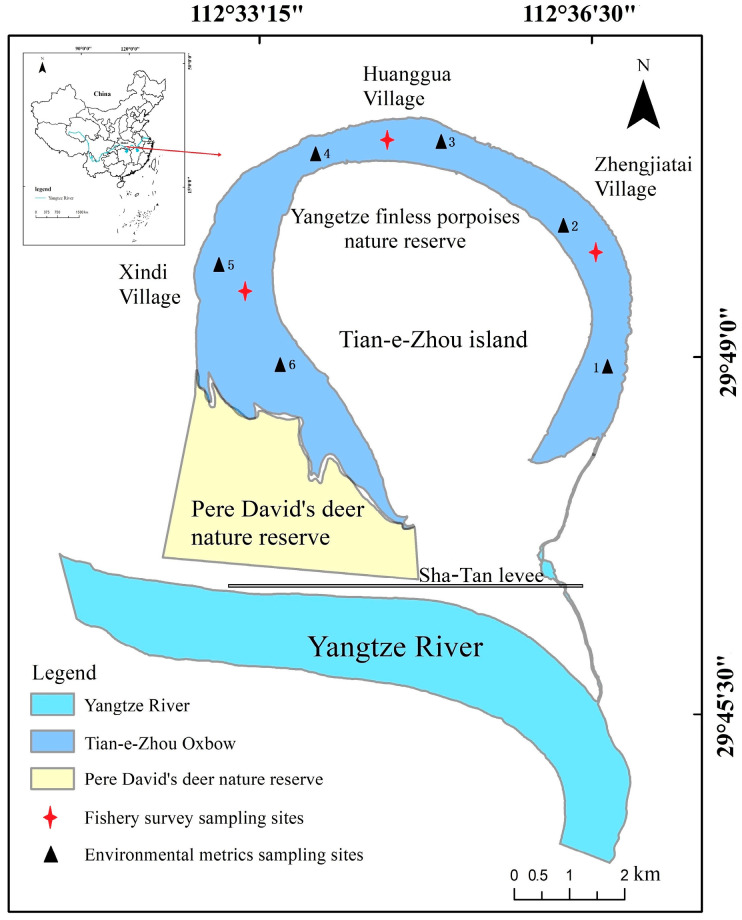
Location of Tian-e-Zhou Oxbow and its sampling sites.

**Figure 2 biology-12-01359-f002:**
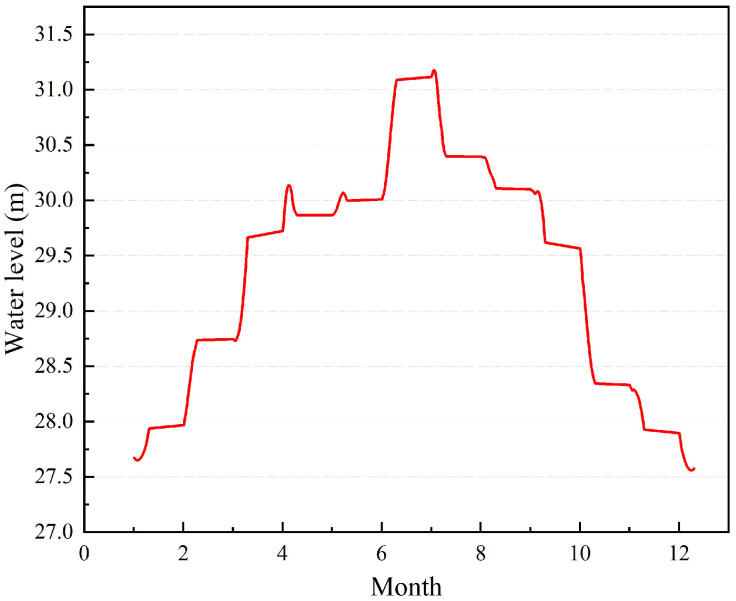
Water-level fluctuations in the waters near Huanggua Village in the Tian-e-Zhou Oxbow in 2021.

**Figure 3 biology-12-01359-f003:**
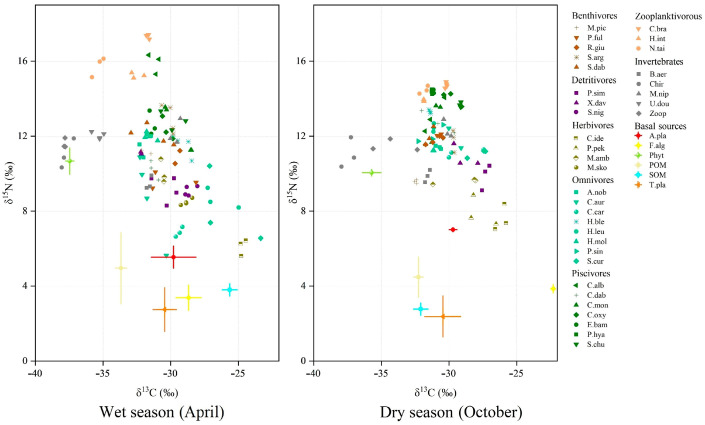
Stable carbon and nitrogen isotope signatures (mean ± SD) of the consumer and basal food sources in the Tian-e-Zhou Oxbow. Symbols of the same color represent fish with similar food habits, and different colors and shapes represent different consumers and food sources. M. pic: *Mylopharyngodon piceus* (*n* = 6); P. ful: *Pelteobagrus fulvidraco* (*n* = 6); R. giu: *Rhinogobius giurinus* (*n* = 6); S. arg: *Squalidus argentatus* (*n* = 7); S. dab: *Saurogobio dabryi* (*n* = 7); P. sim: *Pseudobrama simoni* (*n* = 7); X. dav: *Xenocypris davidi* (*n* = 7); S. nig: *Sarcocheilichthys nigripinnis* (*n* = 2); C. ide: *Ctenopharyngodon idellus* (*n* = 6); P. pek: *Parabramis pekinensis* (*n* = 3); M. amb: *Megalobrama amblycephala* (*n* = 6); M. sko: *Megalobrama skolkovii* (*n* = 4); A. nob: *Aristichthys nobilis* (*n* = 7); C. aur: *Carassius auratus* (*n* = 7); H. ble: *Hemiculter bleekeri* (*n* = 7); H. leu: *Hemiculter leucisculus* (*n* = 6); H. mol: *Hypophthalmichthys molitrix* (*n* = 7); P. sin: *Pseudolaubuca sinensis* (*n* = 7); S. cur: *Squaliobarbus curriculus* (*n* = 6); C. alb: *Culter alburnus* (*n* = 6); C. dab: *Culter dabryi* (*n* = 6); C. mon: *Culter mongolicus* (*n* = 7); C. oxy: *Culter oxycephaloides* (*n* = 6); E. bam: *Elopichthys bambusa* (*n* = 6); P. hya: *P. hyalocranius* (*n* = 3); S. chu: *S. chuats* (*n* = 6); C. bra: *C. brachygnathus* (*n* = 7); H. int: *H. intermedius* (*n* = 6); N. tai: *N. taihuensis* (*n* = 6); B. aer: *Bellamya aeruginosa* (*n* = 6); Chir: Chironomidae (*n* = 6); M. nip: *Macrobranchium nipponense* (*n* = 6); U. dou: *Unio douglasiae* (*n* = 4); Zoop: Zooplankon (n = 6); A. pla: Aquatic vascular plants (*n* = 9); F. alg: filamentous green algae (*n* = 6); Phyt: phytoplankton (*n* = 6); POM: particulate organic matter (*n* = 6); SOM: sedimentary organic matter (*n* = 6); T. pla: terrestrial plants (*n* = 18).

**Figure 4 biology-12-01359-f004:**
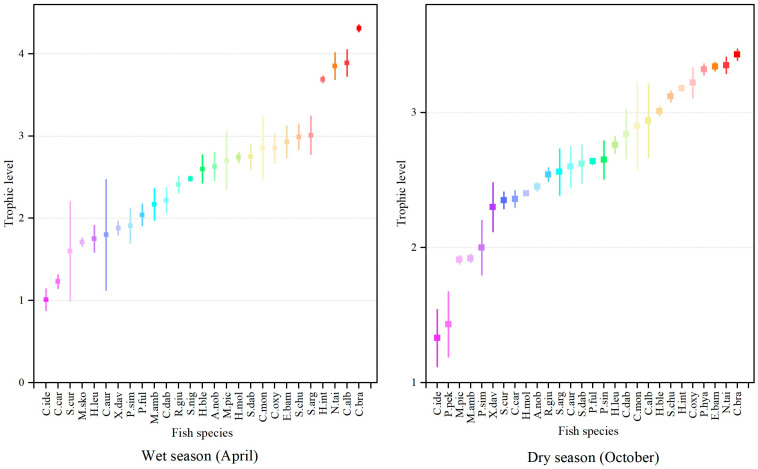
Trophic level distribution (Mean ± SD) characteristics of fish species in the Tian-e-Zhou Oxbow of Yangtze River. The different colors represent different species of fish. (See Figure 1 for abbreviations).

**Figure 5 biology-12-01359-f005:**
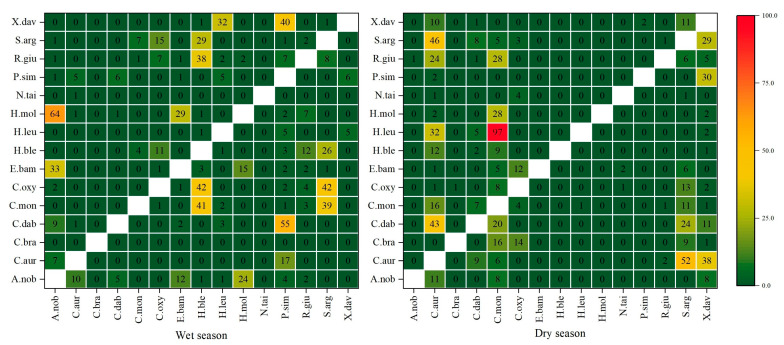
The proportion (%) of SEAc overlap between fish species in the Tian-e-Zhou Oxbow of the Yangtze River during the dry season and the wet season. Nutritional niche overlap contains two directions [59,62]; the number in the figure means the niche overlap percentage of longitudinal species to horizontal species.

**Figure 6 biology-12-01359-f006:**
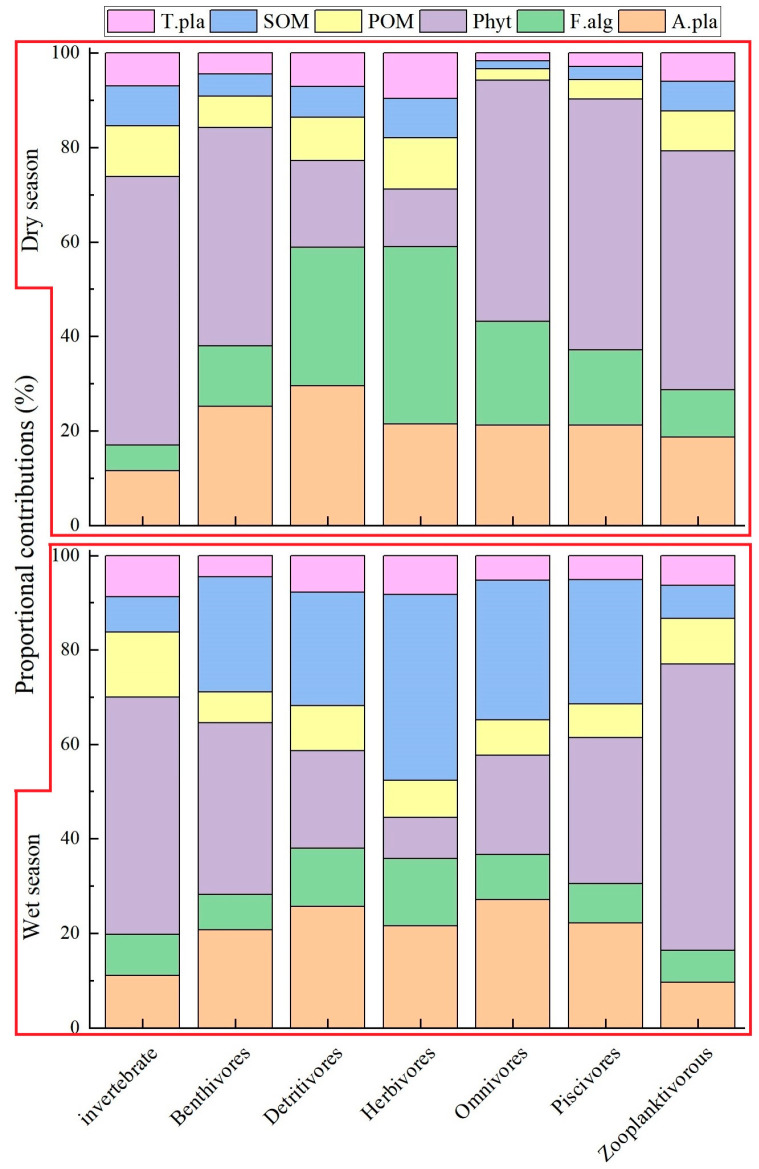
Contributions (mean) of basal resources to consumers in dry and wet seasons in the Tian-e-Zhou Oxbow of Yangtze River. T. pla: terrestrial plants; POM: particulate organic matter; SOM: sedimentary organic matter; Phyt: phytoplankton; F. alg: filamentous green algae; A. pla: aquatic vascular plants.

**Table 1 biology-12-01359-t001:** Physico-chemical characteristics (mean ± SD) in the Tian-e-Zhou Oxbow of the Yangtze River.

Parameters	April (*n* = 5)	July (*n* = 6)	October (*n* = 6)	December (*n* = 6)	One-Way ANOVA
WT (°C)	18.0 ± 0.8 ^b^	28.5 ± 0.2 ^a^	18.1 ± 0.9 ^b^	16.1 ± 0.5 ^c^	*p* = 0.000
DO	10.35 ± 0.43 ^a^	9.72 ± 0.53 ^ab^	8.83 ± 0.74 ^bc^	8.15 ± 0.69 ^c^	*p* = 0.000
SD (m)	76.6 ± 20.2 ^Aa^	68.7 ± 16.2 ^a^	63.5 ± 15.5 ^ab^	41.7 ± 16.7 ^b^	*p* = 0.033
pH	8.85 ± 0.05 ^a^	8.31 ± 0.41 ^b^	8.33 ± 0.08 ^b^	8.19 ± 0.16 ^b^	*p* = 0.002
TDS (mg/L)	362.6 ± 3.4 ^b^	333.0 ± 6.4 ^c^	344.5 ± 7.9 ^c^	305.7 ± 23.6 ^a^	*p* = 0.000
COD_Mn_ (mg/L)	2.21 ± 0.44 ^c^	3.33 ± 0.86 ^ab^	4.08 ± 0.74 ^a^	3.03 ± 0.13 ^cb^	*p* = 0.002
TN (mg/L)	0.60 ± 0.11 ^a^	0.65 ± 0.15 ^a^	0.38 ± 0.08 ^b^	0.63 ± 0.03 ^a^	*p* = 0.002
TP (mg/L)	0.05 ± 0.01 ^a^	0.08 ± 0.04 ^a^	0.04 ± 0.01 ^a^	0.06 ± 0.01 ^a^	*p* = 0.067

*n*: the number of sampling sites. WT: water temperature. DO: dissolved oxygen. SD: Secchi depth. TDS: total dissolved solids. COD_Mn_: the permanganate index. TN: total nitrogen. TP: total phosphorus. Least Significant Difference (LSD), one-way ANOVA, and Duncan’s method were employed for multiple comparisons. Values bearing the different letters demonstrated a significant difference between months (*p* < 0.05), while the same letters demonstrated no significant difference (*p* > 0.05).

**Table 2 biology-12-01359-t002:** Trophic niche metrics of fish in the Tian-e-Zhou Oxbow of the Yangtze River.

Period	Metrics	Piscivores	Detritivores	Omnivores	Herbivores	Zooplanktivores	Benthivores	Total
Wet season	TA	5.62	1.47	17.94	1.23	3.15	9.14	71.58
	SEAc	5.78	3.33	14.71	4.13	5.21	9.92	14.37
	CD	1.51	1.58	2.92	2.65	1.74	1.90	2.89
	MNND	1.45	2.00	1.55	3.16	2.45	2.07	0.89
	SDNND	1.14	0.62	1.15	1.42	0.35	0.88	0.62
	CR	3.09	4.18	8.95	6.28	4.24	4.79	12.47
	NR	6.67	2.86	5.54	5.17	2.32	4.42	11.81
Dry season	TA	1.96	——	3.60	1.08	0.45	2.37	31.49
	SEAc	2.53	2.48	3.09	5.00	0.75	2.42	7.95
	CD	0.96	0.95	1.26	1.36	0.80	1.09	1.99
	MNND	0.34	1.90	0.76	1.82	0.92	0.99	0.59
	SDNND	0.26	——	0.36	0.31	0.60	0.90	0.52
	CR	2.88	2.16	4.96	5.29	2.00	2.92	6.76
	NR	2.82	2.49	2.58	2.39	1.06	2.47	7.87

TA: total area of the convex hull; SEAc: corrected standard ellipse area; CD: mean distance to the centroid; MNND: mean nearest neighbor distance; SDNND: standard deviation of nearest neighbor distance; CR: the δ^13^C range; NR: the δ^15^N range.

**Table 3 biology-12-01359-t003:** Trophic niche parameters of fish in different water bodies in the Yangtze River basin.

Waterbody Name	Times	CR	NR	TA	SEAc	CD	MNND	SDNND	Number of Fish Species
Tian-e-Zhou Oxbow	April (2021)	12.47	11.81	71.58	14.37	2.89	0.89	0.62	27
	October (2021)	6.76	7.78	31.49	7.95	1.99	0.59	0.52	28
Mituo Town of Upper Yangtze River [83]	August (2019)	11.72	9.75	60.67	12.53	2.41	0.59	0.72	23
	November (2019)	8.92	7.67	38.19	8.55	1.98	0.57	0.58	13
Three Gorges Reservoir [82]	2013	8.47	4.34	13.62	--	3.35	1.90	0.63	19
Yangcheng Lake [84]	April (2018)	6.84	6.64	37.07	8.37	1.94	0.56	0.52	34
	October (2017)	6.70	5.72	23.20	7.83	1.96	0.51	0.50	25
Daning River [85]	May (2011)	4.93	7.45	21.41	--	2.37	1.62	0.88	32
	October (2011)	7.45	10.40	38.28	--	3.28	2.22	1.45	23

TA: total area of the convex hull; SEAc: corrected standard ellipse area; CD: mean distance to the centroid; MNND: mean nearest neighbor distance; SDNND: standard deviation of nearest neighbor distance; CR: the δ^13^C range; NR: the δ^15^N range.

## Data Availability

The remaining data presented in this study are available upon reasonable request from the corresponding author. The data are not publicly available due to they involve other unpublished studies.

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
