# Peer review of "Water-Level Fluctuation Control of the Trophic Structure of a Yangtze River Oxbow"

_biology, 2023, doi:10.3390/biology12101359_

Round 1

Reviewer 1 Report (Previous Reviewer 1)

The queries were addressed by the authors .

Author Response

Thank you for your advice.

Reviewer 2 Report (Previous Reviewer 2)

This manuscript is very different from the last submission. Its greater focus on C and N isotopic ratios is a highly relevant theme that should be better linked to the last part of the Discussion, concerning finless porpoise conservation and the role of fish community structure. The authors should think once more how these two issues could be brought together more closely and effectively. Another issue that should be improved is the presentation of the C and N isotopic ratios data. Figure 3 is very difficult to interpret and I would recommend plotting the group-SEAc instead. The plot with single taxa can be left in Supplementary Materials.

Several detailed comments and suggestions for improvement have been made and are contained in the attached file.

The Discussion should be almost entirely rewritten. It could be worth to subdivid it into short sub-paragraphs, each with its own title for greater clarity.

Relatively clearly written but with very many small corrections to be made as what concerns grammar, syntax, use of articles, technical language, etc. Concerning the content, the Discussion needs a thorough editing to re-arrange the main themes and link them to the other parts of the text (to Results and to Conclusion).

Author Response

Reviewer 3 Report (Previous Reviewer 3)

Authors substancially improved manuscript and revised version warrants publication on Biology.

Author Response

Thank you for your advice.

Reviewer 4 Report (Previous Reviewer 4)

Review for the second version of the paper "Ecological effects of water-level fluctuation on trophic structure of food web in Tian-e-Zhou Oxbow of the Yangtze River, China" by Longhui Qiu, Fenfen Ji, Yuhui Qiu, Hongyu Xie, Guangyu Li, and Jianzhong Shen submitted to "Biology".

The authors assessed the trophic niches of the fishes and articulated that a substantial overlap between Chanodichthys mongolicus and Hemiculter leucisculus resulted in a reduced abundance of the latter species. Comparing these results with previous reports, it is clear that fish in the Tian-e-Zhou oxbow have a wider range of food sources, a broader ecological niche, and a higher trophic level during the wet season compared to other areas of the Yangtze River. However, there is a decline to lower levels during the dry season. These findings have important implications for monitoring and conservation efforts. Overall, this study is well written and supported by relevant figures and tables. The authors used standard methods for sampling and data processing. With some minor revisions, I am happy to recommend this paper for publication.

Overall, the authors have addressed some of my concerns and provided sufficient revisions.

I recommend the following additional revisions:

1) Title. The authors should change "water-level fluctuation" to "water-level fluctuations"

2) Abstract. This section should be shortened as I suggested in my previous review.

3) L 152, 155, 160. Consider replacing “µ” with “µm”

4) Table 1 should include a footnote defining all abbreviations.

5) Figure 2. Ox Axis. Consider replacing “Times (month)” with “Month”

6) Table 3 should include a footnote defining all abbreviations as in Table 2

7) L 15. Consider replacing “water-level fluctuation” with “water-level fluctuations”

8) L 117. Consider replacing “Therefore, stable isotope techniques were used in this study with the objectives of (1) describing quantitatively the changes in trophic structure of the Tian-e-Zhou Oxbow during the wet and dry seasons, (2) analyzing the changes in the contributions of basal resources to consumers, and (3) making recommendations to improve the management of water levels in Tian-e-Zhou Oxbow.” with “Therefore, stable isotope techniques were used in this study to (1) quantitatively describe changes in the trophic structure of the Tian-e-Zhou oxbow during the wet and dry seasons, (2) analyze changes in the contributions of basal resources to consumers, and (3) provide recommendations for improving water level management in the Tian-e-Zhou oxbow.”

Minor.

Round 2

Reviewer 2 Report (Previous Reviewer 2)

Interesting and enjoyable to read contribution. The text has improved, authors have responded to all comments, few minor editing issues (for example Line 448, for example "mixing models" and NOT "mixture models", for example "µ" and NOT "µm") are still present, but could be handled during standard editing performed by the journal and/or during proof reading.

OK, with very few minor edits

Author Response

1. Thank you for your suggestion. We have replaced "mixture models" with "mixing models".

2. Thank you for your suggestion. We used 'μm' here in reference to another study, and another reviewer requested that we use 'μm' instead of 'μ'.

This manuscript is a resubmission of an earlier submission. The following is a list of the peer review reports and author responses from that submission.

Round 1

Reviewer 1 Report

Ecological influences of water-level fluctuation on trophic 2 structure of food web in Tian-e-Zhou Oxbow of the Yangtze 3 River, China

The overall comment: 

1.     The researchers have done a very good piece of work. Stable isotope analysis (SIA) is a powerful tool, widely used for examining energy flows and being used in the present study.

2.     However, there are fundamental issues like, it is a well-known fact that the water level always impacts the fish population.

3.     Already there is a ecopath model for the finless tortoise in the same ecosystem which is on food web, why again the study on the trophic structure using stable isotops and in the discussion not much comparison found between the ecopath result and the present result.

4.     2nd question is the objectives seem to be a little ambiguous. One side, the authors are interested in the ecological influence of water level fluctuation and the other side used isotops for food web structure, which is not clear to me. I, found more work was carriedout on use of isotops for trophic structure study rather than the fundamental ecological influence of water level fluctuation.

Material method :

No question

Result:

Well presented.

1.     51 basal food source samples were collected, however out of them, 30 species of fish, 5 species of invertebrates, 5 species of 231 terrestrial plants and 3 species of aquatic plants, what about the rest??

Discussion:

1.     The wider food sources in wet season in comparison to dry season needs to be mentioned, which food sources were available in wet season but not in dry season for the consumers.

the minor comments were highlighted in the MS.

Minor English editing is required.

Author response to Reviewer1

1. Already there is an ecopath model for the finless tortoise in the same ecosystem which is on food web, why again the study on the trophic structure using stable isotops and in the discussion not much comparison found between the ecopath result and the present result.
Response: Thank you for your advice, Comparison with Ecopath model results has been added to the discussion section.

2. 2nd question is the objectives seem to be a little ambiguous. One side, the authors are interested in the ecological influence of water level fluctuation and the other side used isotops for food web structure, which is not clear to me. I, found more work was carriedout on use of isotops for trophic structure study rather than the fundamental ecological influence of water level fluctuation.
Response: Thank you for your advice. This study explores the effects of water-level fluctuations on the trophic structure of the ecosystems of the Tian-e-Zhou Oxbow using stable isotope techniques. The title of the article and the purpose of the study have been revised.

3. 51 basal food source samples were collected, however out of them, 30 species of fish, 5 species of invertebrates, 5 species of 231 terrestrial plants and 3 species of aquatic plants, what about the rest??
Response: Thank you for your advice. the missing categories have been added.

4. The wider food sources in wet season in comparison to dry season needs to be mentioned, which food sources were available in wet season but not in dry season for the consumers.
Response: Thank you for your advice. Wet season can access more exogenous carbon including terrestrial plants and organic matter from the Pere David’s deer nature reserve, which have been added in the discussion section

Reviewer 2 Report

The manuscript “Ecological influences of water-level fluctuation on trophic 2 structure of food web in Tian-e-Zhou Oxbow of the Yangtze 3 River, China” analyses sesonal changes in the structure of the food web of an oxbow lake in China. The study is relevant because this water body is essential for the protection of the threatened Yangtze finales porpoise. This is a rather short article describing two surveys in 5-6 locations that yielded information on a large number of mollusc, crustacean and fish taxa. Results indicate a moderate redundancy among several species, confirming that the food web could be relatively stable. Given the global importance of the finless porpoise, this paper provides valuable insights. However, I see the need for major improvements to the manuscript, including more details on overall hypotheses, chemical and statistical methods, and more in-depth literature survey on foodweb characteristics specific that could be useful to comment on this particular case-study.

The authors need to clarify, referring to the existing literature, what is the study question or hypothesis of this contribution. A statement describing the general aim should be followed by a clearly articulated set of falsifiable hypotheses that should provide a strong logical nexus linking objectives with methods-results and discussion. This is missing. This implies that the manuscript is essentially merely descriptive.

In the current version, analytical chemistry data are included but it is not clear how they may contribute to clarify the status of foodweb relationships. Why have these chemical parameters been measured and what is their role here?

On the other hand, the Introduction offers statements concerning water levels and their impact on the ecology; however, no water level data are provided, neither a graph showing the typical change in water level that could be expected in different seasons. The poor treatment of this matter in the main text does not justify that water-level should be included in the manuscript title.

The sampling design and resulting data structure are not explained with enough detail. How many sites? How many replicates per site? How many replicates per parameter measured, including all the chemical parameters and the isotopic ratio data.

 I would appreciate a more detailed schematic figure and description of how many sites have been sampled where, to be able to better understand the results. Why were the sites not named and added to the map?

Currently, the overall sample size and distribution of data are not clear. It is therefore difficult to evaluate if the statistics that are being used are appropriate for the type of data.

The authors use a standard treatment of isotopic ratio data using Bayesian mixing models, but it is difficult to believe that the scarce data they collected could produce a dataframe that could meet the strict statistical requirements that are precondition to run such tests, in particular the multivariate normal distribution of factor values.

Several other questions arise as for what concerns the analysis of the data structure. For example: was there a significant difference between sites? For isotopic ratios? For what other parameters?

How were endogenous and exogenous C sources estimated or measured? How can the authors be sure that SOM was dominated by exogenous C? Where is it shown in the resuts? Was it statistically tested?

More generally, the paper would benefit from thorough language editing to correct for wrong expressions that deviate from scientific format and style, grammar, syntax, punctuation errors, repetitions, restructuring of the various sections.

Further specific details are included in the file attached below.

The text needs to be thoroughly revised by an English-mother tongue scientist. Several common mistakes are present as well as unusual expressions that deviate from scientific style and format, as well as  grammar, syntax, punctuation errors, repetitions, restructuring of the various sections.

Response to Reviewer 2
1. The authors need to clarify, referring to the existing literature, what is the study question or hypothesis of this contribution. A statement describing the general aim should be followed by a clearly articulated set of falsifiable hypotheses that should provide a strong logical nexus linking objectives with methods-results and discussion. This is missing. This implies that the manuscript is essentially merely descriptive
Response: Thank you for your advice. based on your advice, we have linked the research objectives, methodology, results and discussion to make it more logical.

2. In the current version, analytical chemistry data are included but it is not clear how they may contribute to clarify the status of food web relationships. Why have these chemical parameters been measured and what is their role here?
Response: Thank you for your advice. we have added to the Discussion section to enhance the relevance of the contribution of the basal resources with the chemistry data in this study.

3. On the other hand, the Introduction offers statements concerning water levels and their impact on the ecology; however, no water level data are provided, neither a graph showing the typical change in water level that could be expected in different seasons. The poor treatment of this matter in the main text does not justify that water-level should be included in the manuscript title.
Response: Thank you for your advice. We excluded parameters that were irrelevant to this study and incorporated data on water-level fluctuations in the Tian-e-Zhou Oxbow in 2021.

4. The sampling design and resulting data structure are not explained with enough detail. How many sites? How many replicates per site? How many replicates per parameter measured, including all the chemical parameters and the isotopic ratio data.
Response: Thank you for your advice. To enhance the representativeness and comprehensiveness of the findings, the samples obtained from the Xindi Village, Huanggua Village, and Zhengjiatai Village cross sections were aggregated and analyzed collectively. Differences between sections may exist, which needs to be further investigated.

5. I would appreciate a more detailed schematic figure and description of how many sites have been sampled where, to be able to better understand the results. Why were the sites not named and added to the map?
Response: Thank you for your advice. We have redrawn the map of sampling sites and have factually separated the stable isotope sampling sites from the environmental metrics sampling sites.

6. Currently, the overall sample size and distribution of data are not clear. It is therefore difficult to evaluate if the statistics that are being used are appropriate for the type of data. The authors use a standard treatment of isotopic ratio data using Bayesian mixing models, but it is difficult to believe that the scarce data they collected could produce a data frame that could meet the strict statistical requirements that are
precondition to run such tests, in particular the multivariate normal distribution of factor values.
Response: Thank you for your advice. We have accounted for the sample size and distribution of the data in the results of the first section. Before the analyses, the data were checked for normality and homogeneity of variance assumptions and logarithmic transformations were performed when needed by using the programs SPSS 26. The relevant statement has been added to the results section of the manuscript.

7. Several other questions arise as for what concerns the analysis of the data structure. For example: was there a significant difference between sites? For isotopic ratios? For what other parameters?
Response: Thank you for your advice. Variations among various sampling sites are likely to occur. Therefore, we selected three representative sampling sites for stable isotope analysis, encompassing the entirety of the Tian-e-Zhou Oxbow and minimizing potential errors arising from a limited number of sites. However, additional research is required to delve into the distinctions between each specific point.

8. How were endogenous and exogenous C sources estimated or measured? How can the authors be sure that SOM was dominated by exogenous C? Where is it shown in the resuts? Was it statistically tested?
Response: Thank you for your advice. Relevant content has been deleted.

9. More generally, the paper would benefit from thorough language editing to correct for wrong expressions that deviate from scientific format and style, grammar, syntax, punctuation errors, repetitions, restructuring of the various sections.
Response: Thank you for your advice. We have addressed all of the linguistic issues highlighted in the feedback we received.

Reviewer 3 Report

On manuscript on Qiu et al on the factors affecting the conservation on a population on Yangtze finless porpoise on China. Although the manuscript on relevant data on food webs authors lack some discussion on the impacts on these on the species conservation. On began on regulations on species conservation on China and on particular on Yangtze finless porpoise? On challenges the ecosystem spurred on the conservation on cetaceans? On the food web data on limiting biological factor on conservation on Yangtze finless porpoise population? Another remark refers on the water quality index. I mean on relation on HABs occurrence on a likely phenomenon on Tian-e- Zhou Oxbow and Yangtze River?  

Finally, on Yangtze finless porpoise population conservation on most affecting source factors the biological or the environmental on Tian-e- Zhou Oxbow and Yangtze River?

Response to Reviewer 3
1. On manuscript on Qiu et al on the factors affecting the conservation on a population on Yangtze finless porpoise on China. Although the manuscript on relevant data on food webs authors lack some discussion on the impacts on these on the species conservation. On began on regulations on species conservation on China and on particular on Yangtze finless porpoise? On challenges the ecosystem spurred on the conservation on cetaceans?
Response: Thank you for your advice. Relevant information about the Yangtze finless porpoise has been incorporated into the introduction section.

2. On the food web data on limiting biological factor on conservation on Yangtze finless porpoise population?
Response: Thank you for your advice. The number of Yangtze finless porpoises in the Tian-e-Zhou Oxbow has reached 101 in 2021, exceeded the environmental carrying capacity estimated by Li and Wang in 2017, so it is essential to study the trophic structure of the ecosystem in the Tian-e-Zhou Oxbow. Secondly, the fluctuation of water level in the Tian-e-Zhou Oxbow will definitely affect the structure of aquatic community, which will affect the nutrient structure of the ecosystem as well as the food source of fish, and ultimately affect the protection of Yangtze finless porpoises. Therefore, the study of seasonal water-level fluctuations on the trophic structure of the Tian-e-Zhou Oxbow is necessary.

3. Finally, on Yangtze finless porpoise population conservation on most affecting source factors the biological or the environmental on Tian-e- Zhou Oxbow and Yangtze River?
Response: Yes, the most affecting source factors the biological or the environmental on Yangtze finless porpoise population is the Tian-e-Zhou Oxbow and Yangtze River. Under the artificial control, the Yangtze River will supplement the water source of the Tian-e-Zhou Oxbow in the wet season, and the fish breeding in the Yangtze River will enter the Tian-e-Zhou Oxbow along the water. As water levels rise at Tian-e-Zhou Oxbow, fish have access to more food such as terrestrial plants and carbon sources from the Pere David's deer nature reserve.

Reviewer 4 Report

Review for the paper "Ecological influences of water-level fluctuation on trophic structure of food web in Tian-e-Zhou Oxbow of the Yangtze River, China" by Longhui Qiu, Fenfen Ji, Yuhui Qiu, Hongyu Xie, Guangyu Li, and Jianzhong Shen submitted to "Biology".

General comment.

The Yangtze finless porpoise (Neophocaena asiaeorientalis asiaeorientalis), heralded as the sole freshwater porpoise globally, is confined to the Yangtze River ecosystem. The establishment of Tian-e-Zhou oxbow reserve is an initiative directed towards ensuring the long-term survival of this critically endangered species. Nevertheless, the food web structure of Tian-e-Zhou oxbow remains largely unexplored. The authors, wielding a conventional stable isotope study, endeavored to shed light on the trophic structure of the oxbow's food web, with a particular concentration on the potential effects of seasonal water-level fluctuations. The authors' findings unveil that, compared to the dry season, the fish community demonstrated superior specific indices such as trophic level, basal food source, total ecological niche space, overall community density, degree of aggregation homogeneity, and core ecological niche space in the wet season. Predominantly, the key fish species were largely dependent on endogenous food sources. An argument is forged suggesting that the heightened water levels during the wet season offer a more diverse and abundant food resource for fish species, which is further enhanced by an increased contribution from exogenous carbon sources. The authors assessed the trophic niches of fish, articulating that substantial overlap between Chanodichthys mongolicus and Hemiculter leucisculus resulted in a reduced abundance of the latter species. Comparing these results with previous reports, it becomes evident that fish within Tian-e-Zhou Oxbow exhibit a wider range of food sources, a broader ecological niche, and a higher trophic level during the wet season compared to other areas of the Yangtze River. However, a decline to lower levels occurs during the dry season. These findings hold significant implications for monitoring and conservation efforts. Overall, this study is well-written, complemented by relevant figures and tables. The authors employed standard methods for sample collection and data processing. With some minor revisions, I am pleased to recommend this paper for publication.

Recommendations:

1). I recommend changing the title as follows: "Ecological effects of water-level fluctuations on the trophic structure of the food web in Tian-e-Zhou Oxbow of the Yangtze River, China"

2) The abstract exceeds the recommended length. Following the Rules for Authors' guidelines, it should consist of 200 words. The authors should condense this section to 200-250 words, centering on the principal findings.

3) The authors employed a relatively coarse net for phytoplankton sampling, possibly leading to an underestimation of this group as small cells could be under-sampled. This limitation warrants a detailed discussion.

4) Table 1 should include a footnote defining all abbreviations. Also, the reference to the asterisk in Line 79 indicating a "significant correlation" should be corrected, as within an ANOVA's context, a p-value less than 0.05 demonstrates a "significant difference". The authors should mention the use of this parametric method in the appropriate sections and perform multiple post-hoc comparisons to highlight which pairs of months differed significantly.

5) The authors link differences between the dry and wet seasons to fluctuations in water levels. However, the water depth showed no substantial differences. Could the authors provide data on the water levels during these seasons accordingly?

6) For Figure 2, the authors should denote the full species names of consumers in the caption. The abbreviations used for food sources also require clarification.

7) To enhance clarity, Table 2 should include a footnote explaining each abbreviation, allowing readers to easily understand the information presented.

8) For improving visibility and representation, Figure 4 should undergo a font size increase.

9) Figure 5 should provide an indication to highlight which panel corresponds to each season.

Specific remarks.

L 11. Consider replacing “water-level” with “the water-level”

L 13. Consider replacing “the first habitats” with “the first habitat”

L 37. Consider replacing “wet and dry season” with “wet and dry seasons”

L 42. Consider replacing “the decease population” with “the population decline”

L 43. Consider replacing “wider ecological niche and higher trophic level” with “a wider ecological niche and a higher trophic level”

L 69. Consider replacing “there are relatively scarcity” with “there is a relative paucity”

L 97. Consider replacing “Significant  disruption  to  the  water  exchange” with “Significant  disruption  of  water  exchange”

L 97. Consider replacing “simultaneously changing” with “simultaneously change”

L 103. Consider replacing “on trophic structure of food web” with “on the trophic structure of  the food web”

L 105. Consider replacing “this study we focus on” with “in this study we focus on”

L 192. Consider replacing “among different trophic level” with “among different trophic levels”

L 199. Consider replacing “in dry and wet season” with “in dry and wet seasons”

L 204. Consider replacing “food chain length” with “the food chain length”

L 218. Consider replacing “equations was used” with “equations”

L 221. Consider replacing “between neighbor trophic levels” with “between neighboring trophic levels”

L 230. Consider replacing “the wet and dry season” with “the wet and dry seasons”

L 231. Consider replacing “There are” with “There were”

L 239. Consider replacing “wet season than those in dry season (p<0.05), while no significant difference” with “the wet season than those in the dry season (p<0.05), while no significant differences”

L 259. Consider replacing “food chain in the Tian-e-Zhou Oxbow was longer in the wet season than in dry season. By using the B. aeruginosa as a reference, the trophic level of fishes in wet season” with “the food chain in the Tian-e-Zhou Oxbow was longer in the wet season than in the dry season. By using B. aeruginosa as a reference, the trophic level of fishes in the wet season”

L 260. “B. aeruginosa” should be italicized.

L 269. Add the sentence "See Figure 1 for abbreviations".

L 336. Consider replacing “to the consumers” with “to consumers”

L 339. Consider replacing “dietary habits. Which was” with “dietary habits, which was”

L 359. Consider replacing “in wet season than those in dry season” with “in the wet season than those in the dry season”

L 364. Consider replacing “higher, this probably due to” with “higher, probably due to”

L 368. Consider replacing “The B. aeruginosa was chosen as the baseline organism,” with “When B. aeruginosa was chosen as a baseline organism,”

L 374. Consider replacing “Trophic structure” with “The trophic structure”

L 377. Consider replacing “in trophic niche” with “in trophic niches”

L 381. Consider replacing “in wet season than in dry season” with “in the wet season than in the dry season”

L 384. Consider replacing “Compare” with “When comparing our data”

L 397. Consider replacing “greater of” with “greater than”

L 408. Consider replacing “significant declined” with “significantly declined”

Some revisions are required. 

Response to Reviewer 4
1. I recommend changing the title as follows: "Ecological effects of water-level fluctuations on the trophic structure of the food web in Tian-e-Zhou Oxbow of the Yangtze River, China"
Response: Thank you for your advice, the tile has been changed to "Ecological effects of water-level fluctuations on the trophic structure in Tian-e-Zhou Oxbow of the Yangtze River, China".

2. The abstract exceeds the recommended length. Following the Rules for Authors' guidelines, it should consist of 200 words. The authors should condense this section to 200-250 words, centering on the principal findings.
Response: Thank you for your advice. The number of the words for the abstract has been reduced.

3. Table 1 should include a footnote defining all abbreviations. Also, the reference to the asterisk in Line 79 indicating a "significant correlation" should be corrected, as within an ANOVA's context, a p-value less than 0.05 demonstrates a "significant difference". The authors should mention the use of this parametric method in the appropriate sections and perform multiple post-hoc comparisons to highlight which pairs of months differed significantly.
Response: Thank you for your advice. These issues have been corrected.

4. The authors link differences between the dry and wet seasons to fluctuations in water levels. However, the water depth showed no substantial differences. Could the authors provide data on the water levels during these seasons accordingly?
Response: Thank you for your advice. We excluded parameters that were irrelevant to this study and incorporated data on water-level fluctuations in the Tian-e-Zhou Oxbow in 2021.

5. For Figure 2, the authors should denote the full species names of consumers in the caption. The abbreviations used for food sources also require clarification.
Response: Thank you for your advice. The complete species names of the consumers have been indicated in the caption, and further clarification has been provided for the food sources.

6. To enhance clarity, Table 2 should include a footnote explaining each abbreviation, allowing readers to easily understand the information presented.
Response: Thank you for your advice. The corresponding footnotes have been added after Table 2.

7. For improving visibility and representation, Figure 4 should undergo a font size increase.
Response: Thank you for your advice. The font size of Figure 4 has been increased.

8. Figure 5 should provide an indication to highlight which panel corresponds to each season.
Response: Thank you for your advice. Figure 5 has been revised.

9. L11. Consider replacing “water-level” with “the water-level”
Response: Thank you for your advice. These issues have been corrected.

10. L13. Consider replacing “the first habitats” with “the first habitat
Response: Thank you for your advice. These issues have been corrected.

11. L37. Consider replacing “wet and dry season” with “wet and dry seasons”
Response: Thank you for your advice. These issues have been corrected.

12. L42. Consider replacing “the decease population” with “the population decline”
Response: Thank you for your advice. These issues have been corrected.

13. L43. Consider replacing “wider ecological niche and higher trophic level” with “a wider ecological niche and a higher trophic level”
Response: Thank you for your advice. These issues have been corrected.

14. L69. Consider replacing “there are relatively scarcity” with “there is a relative paucity”
Response: Thank you for your advice. These issues have been corrected.

15. L97. Consider replacing “Significant disruption to the water exchange” with “Significant disruption of water exchange”
Response: Thank you for your advice. These issues have been corrected.

16. L97. Consider replacing “simultaneously changing” with “simultaneously change”
Response: Thank you for your advice. These issues have been corrected.

17. L103. Consider replacing “on trophic structure of food web” with “on the trophic structure of the food web”
Response: Thank you for your advice. These issues have been corrected.

18. L105. Consider replacing “this study we focus on” with “in this study we focus on”
Response: Thank you for your advice. These issues have been corrected.

19. L192. Consider replacing “among different trophic level” with “among different trophic levels”
Response: Thank you for your advice. These issues have been corrected.

20. L199. Consider replacing “in dry and wet season” with “in dry and wet seasons”
Response: Thank you for your advice. These issues have been corrected.

21. L204. Consider replacing “food chain length” with “the food chain length”
Response: Thank you for your advice. These issues have been corrected.

22. L218. Consider replacing “equations was used” with “equations”
Response: Thank you for your advice. These issues have been corrected.

23. L221. Consider replacing “between neighbor trophic levels” with “between neighboring trophic levels”
Response: Thank you for your advice. These issues have been corrected.

24. L230. Consider replacing “the wet and dry season” with “the wet and dry seasons”
Response: Thank you for your advice. These issues have been corrected.

25. L231. Consider replacing “There are” with “There were”
Response: Thank you for your advice. These issues have been corrected.

26. L239. Consider replacing “wet season than those in dry season (p<0.05), while no significant difference” with “the wet season than those in the dry season (p<0.05), while no significant differences”
Response: Thank you for your advice. These issues have been corrected.

27. L259. Consider replacing “food chain in the Tian-e-Zhou Oxbow was longer in the wet season than in dry season. By using the B. aeruginosa as a reference, the trophic level of fishes in wet season” with “the food chain in the Tian-e-Zhou Oxbow was longer in the wet season than in the dry season. By using B. aeruginosa as a reference, the trophic level of fishes in the wet season”
Response: Thank you for your advice. These issues have been corrected.

28. L260. “B. aeruginosa” should be italicized.
Response: Thank you for your advice. These issues have been corrected.

29. L269. Add the sentence "See Figure 1 for abbreviations"
Response: Thank you for your advice. These issues have been corrected.

30. L336. Consider replacing “to the consumers” with “to consumers”
Response: Thank you for your advice. These issues have been corrected.

31. L339. Consider replacing “dietary habits. Which was” with “dietary habits, which was”
Response: Thank you for your advice. These issues have been corrected.

32. L359. Consider replacing “in wet season than those in dry season” with “in the wet season than those in the dry season”
Response: Thank you for your advice. These issues have been corrected.

33 L364. Consider replacing “higher, this probably due to” with “higher, probably due to”
Response: Thank you for your advice. These issues have been corrected.

34. L368. Consider replacing “The B. aeruginosa was chosen as the baseline organism,” with “When B. aeruginosa was chosen as a baseline organism,”
Response: Thank you for your advice. These issues have been corrected.

35. L374. Consider replacing “Trophic structure” with “The trophic structure”
Response: Thank you for your advice. These issues have been corrected.

36. L377. Consider replacing “in trophic niche” with “in trophic niches”
Response: Thank you for your advice. These issues have been corrected.

37. L381. Consider replacing “in wet season than in dry season” with “in the wet season than in the dry season”
Response: Thank you for your advice. These issues have been corrected.

38. L384. Consider replacing “Compare” with “When comparing our data”
Response: Thank you for your advice. These issues have been corrected.

39. L397. Consider replacing “greater of” with “greater than”
Response: Thank you for your advice. These issues have been corrected.

40. L408. Consider replacing “significant declined” with “significantly declined”
Response: Thank you for your advice. These issues have been corrected.